# Internalization of Phospholipid-Coated Gold Nanoparticles

**Lindsay J. Shearer [1,2,*] and Nils O. Petersen [1,2]**

[1]   Department of Chemistry, University of Alberta, Edmonton, AB T6G2G2, Canada; nils.petersen@ualberta.ca
[2]   National Institute for Nanotechnology, National Research Council, Edmonton, AB T6G2M9, Canada
*   Correspondence: lindsay@ualberta.ca

**Abstract:** Gold nanoparticles are used in health-related research; however, their effectiveness appears to depend on how well they are internalized and where they are destined to travel. Internalization in cells is efficient if the gold nanoparticles are biocompatible, where one possible pathway of cell entry and processing is clathrin-mediated endocytosis. In this work we studied the co-localization of phospholipid-coated gold nanoparticles (PCAuNPs) with markers of the endocytic pathway (Rab and LAMP-1 proteins) in C2C12 and A549 cells and found that the internalization was consistent with clathrin-mediated endocytosis and was cell type dependent. We further found that the time evolution of uptake and disposal of these PCAuNPs was similar for both cell types, but aggregation was more significant in A549 cells. Our results support the use of these PCAuNPs as models for potential drug delivery platforms.

**Keywords:** gold nanoparticles; cells; endocytosis; confocal microscopy; fluorescence; image correlation spectroscopy

---

## 1. Introduction

Nanoparticles have shown promise as platforms for curative treatments and detection of cancer and disease [1–3]. Due to the leaky vasculature of solid tumors, nanoparticles can passively penetrate through these openings and reach the outside cells of solid tumors, a process known as the enhanced permeability and retention effect [4,5]. Gold nanoparticles are particularly favorable as they have shown to be used in a broad range of bio-health applications, specifically in photothermal therapy, drug delivery, and photodynamic therapy [6,7]. Gold nanoparticles are also favorable because they possess the unique ability to bind with thiol groups, thus biomolecules can be easily functionalized to gold for various applications [8,9]. In addition, gold nanoparticles are not significantly toxic in cells, based on lactate dehydrogenase (LDH), metabolic cell activity assays (MTT) [10], and other cell-based toxicity assays [11,12]. Thus, gold nanoparticles are useful for drug delivery platforms and there is large motivation for in-vitro studies [13–19].

Studies have shown that gold nanoparticles can be internalized into cells by a process called clathrin-mediated endocytosis [20–22]. When internalized, the nanoparticles are compartmentalized into vesicles that fuse with early endosomes, which in turn mature into late endosomes, and ultimately fuse with acidic lysosomes for possible degradation [23–27]. Additionally, receptors are recycled back to the cell membrane via recycling endosomes that "pinch" off from the early endosome [28]. Thus, through clathrin-mediated endocytosis, gold nanoparticle processing could serve as a useful platform for cleavage of drugs in the acidic environment of endosomes for cellular processing [29,30]. Endocytosis of any material, like gold nanoparticles, is strongly dependent on membrane trafficking between compartments, and Rab and LAMP proteins are important for assisting with this role [31–33].

Specific Rab and LAMP-1 proteins are known to localize to domains on endocytic compartments; they are referred to as *markers* for these endosomes [34,35]. Rab5 is used as a marker for early endosomes [36–39] and Rab7 is used as a marker for late endosomes [40–43]. Rab11 is used as a marker for recycling endosomes [44–46]. Few, if any Rab proteins have been identified to localize specifically to the lysosome, however the lysosomal associated membrane protein (LAMP-1) is known as a marker for lysosomes [47–49]. In this work, we used antibodies against Rab5, Rab11, Rab7, and LAMP-1 to study their distribution and possible co-localization with gold nanoparticles within the cell. It should be noted that in recent work we have shown that these markers are not as specific for the indicated endosomes as thought in the past [50]. Specifically, we found that Rab5, Rab7, and Lamp1 can all co-exist on the same endosome or lysosomes to varying extents in both C2C12 and A549 cells. Nevertheless, these markers are still clear markers for the endocytic pathway.

Our laboratory created a new type of phospholipid-coated gold nanoparticle (PCAuNP) that can be labeled with fluorescent lipids for easy monitoring with fluorescence microscopy [51]. Subsequently, Wang and Petersen provided a comprehensive characterization of PCAuNPs, in which the structure, stability, aggregation, quenching, FRET, size, and shape of the PCAuNPs were determined [52]. The key properties are described more fully in the materials section below.

Wang and Petersen demonstrated that these PCAuNPs were taken up by A549 cells into acidic compartments [51] but did not characterize which acidic compartments were involved. They determined that they were neither in mitochondria nor in peroxisomes, but by electron microscopy, they were seen in intracellular vesicular compartments, which could be endosomes, and in some cases, they were found in lamellar bodies, structures unique to A549 cells. Accordingly, in this work, we sought to answer the following three questions; (1) Do these PCAuNPs internalize by clathrin-mediated endocytosis, (2) Is this PCAuNP endocytosis cell type dependent, and (3) What happens to these PCAuNPs with time?

Using image cross correlation spectroscopy analysis of laser scanning confocal microscopy images, we explored the uptake of PCAuNPs in two cell types, C2C12 and A549 cells. The A549 cell line was chosen because it is derived from type II alveolar epithelial cells that are known to process lipid coated material, which is why we chose these cells in our previous work [51]. The C2C12 cell line was chosen for comparison with the A549 cells because it is a mouse myoblast cell line that is used to study differentiation and biochemical pathways. Both cells lines are well characterized and widely used [53]. Moreover, they grow as monolayers and are flat, and therefore, they are easy to use in confocal microscopy studies. We first incubated cells with fluorescent PCAuNPs followed by fixation and immunofluorescent labeling of antibodies specific for markers of the clathrin-mediated endocytic pathway. We observed that PCAuNPs were internalized and processed by clathrin-mediated endocytosis with similar uptake pathways in both cell types; however, the extent of aggregation was far greater in A549 cells.

## 2. Materials and Methods

C2C12 and A549 cells were obtained from American Type Culture Collection™. Dulbecco's modified Eagle medium (DMEM), fetal bovine serum (FBS), and 0.25% Trypsin-EDTA were obtained from Invitrogen Life Technologies™. PFA was obtained from EMD®.

Primary antibodies specific for Rab5 and LAMP-1 were obtained from Abcam®. Primary antibodies specific for Rab7 were obtained from Cell Signaling Technology®. All primary antibodies used were monoclonal. Secondary antibodies for Rab7, and LAMP-1 were obtained from Cell Signaling Technology®. Secondary antibody specific for Rab5 was obtained from Abcam®.

Cell culture and sample labelling experiments were done at the National Institute for Nanotechnology (NINT) in Edmonton, Alberta. Fluorescent imaging was done at the Cell Imaging Facility at the Cross Cancer Institute in Edmonton, Alberta.

### 2.1. Cell Culture

Mouse muscle myoblastoma (C2C12) cells were cultured in Dulbecco's modified Eagle medium (DMEM), supplemented with 10% fetal bovine serum (FBS). Human alveolar adenocarcinoma (A549) cells were cultured in Ham's F-12 K medium (F-12 K) supplemented with 10% FBS. The cells were grown in an incubator chamber maintained at 37 °C with a 5% carbon dioxide ($CO_2$) atmosphere, and were passaged every five days using 0.25% Trypsin-EDTA. At 80% cell surface confluence, the cells were passaged 1:5 onto 35 mm glass bottom dishes and maintained in a 37 °C, 5% $CO_2$ incubator for about two days when approximately 60% cell confluency was reached for experimentation. At this time, the media was replaced with 1 mL of fresh media and treated as indicated below.

### 2.2. Phospholipid-Coated Gold Nanoparticle Preparation

Phospholipid-coated gold nanoparticles (PCAuNPs) were prepared by a "three step" seed growth mediated method described previously [51,52]. The first two steps of this method included the preparation of two separate solutions; the growth solution and the seed solution. The PCAuNPs were then prepared by mixing these solutions with a weak reducing agent.

The growth solution was prepared in a 25 mL round bottom flask. First, potassium gold (III) chloride ($KAuCl_4$) was dissolved in water to a final concentration of 0.48 mM. This was stirred in the 25 mL round bottom flask for five minutes. Next, 0.114 g of 1-stearoyl-2-oleoyl-sn-glycero-3-phospho-(1′-rac-glycerol) (sodium salt) (SOPG) phospholipid, was added to the flask. Last, a nitro-benzoxadiazol conjugated SOPG lipid (NBD-SOPG), 1-oleoyl-2-{12-[(7-nitro-2-1,3-benzoxadiazol-4-yl)amino]dodecanoyl}-sn-glycero-3-[phospho-rac-(1-glycerol)] (ammonium salt), was added to the flask at a concentration of 8.8% (*w/w*). The solution was then left to stir for an additional thirty minutes.

The seed solution was prepared in a separate 50 mL round bottom flask. First, $KAuCl_4$ was dissolved in water to a final concentration of 0.64 mM. This was stirred in the flask for five minutes. Second, 10 mL of a 3.8 mM solution of trisodium citrate ($Na_3C_6H_5O_7$) in water was added to the flask and stirred for an additional ten minutes. Last, 3 mL of a 2.9 mM solution of sodium borohydride ($NaBH_4$) in water was added to the flask. The solution was then stirred for five minutes to produce gold "seeds" between 2–3 nanometers (nm) in size.

In the final step, 5 mL of growth solution was first stirred in a separate 25 mL round bottom flask. Second, 0.3 mL of the seed solution was added to the flask and stirred for an additional ten minutes. Last, 0.25 mL of a 9.0 mM solution of ascorbic acid ($C_6H_8O_6$) in water was added to the flask to produce phospholipid-coated gold nanoparticles (PCAuNPs) of approximately 20 nm in size. To ensure monodispersity, the PCAuNPs were stirred for one hour prior to being placed in a 4 °C fridge. The PCAuNPs were maintained in the fridge at 4 °C prior to experimental use.

### 2.3. Characterization and Size Distribution of Phospholipid-Coated Gold Nanoparticles

Phospholipid-coated gold nanoparticles prepared in this manner were fully characterized previously by Wang and Petersen [52]. They reported that comparison of measured and calculated diffraction patterns showed that the phospholipid-coated gold nanoparticles had a face-centred cubic lattice structure (Au (a − b = c = 4.0789, Sys. Cubic, Fm-3m)) and that high-resolution transmission electron microscopy showed gold structures with sharp edges consistent with cubo-octahedral particles that would arise from growth along the 100 surface. They looked either cubic or hexagonal, depending on the orientation at which they were observed. They also found that the synthetic procedure could control the average particle size as determined by electron microscopy and dynamic light scattering. Examples of synthetic procedures to produce 30 nm and 42 nm average sizes were shown along with larger cubic structures of 120–250 nm. They reported three pieces of evidence to support that there is a coating of a phospholipid membrane on these particles: (1) High resolution transmission electron microscopy revealed a 4 nm thick low contrast material on the exterior of the high contrast gold particle (consistent with a 4 nm thick phospholipid bilayer). (2) The zeta-potential was −65 mV, which is more

negatively charged than the citrate-coated gold nanoparticles with about −40mV. This is consistent with a coating of negatively charged phospholipids (here phosphatidyl glycerol (PG)). This larger negative zeta-potential also contributes to the stability of these particles in solution as they are less prone to agglomeration. (3) The plasmon resonance of both 30 nm and 42 nm gold nanoparticles are red-shifted by about 5 nm relative to the resonance measured by Link and El-Sayed [54] for citric acid-coated nanoparticles, which, according to theories by Liz-Marzan et al. [55] and calculations by Van Dijk [56], is consistent with a 4 nm coating of a material with a refractive index of 1.46 (which compares with the refractive index of phospholipids of 1.456). They further found that the fluorescence of the NBD-labeled phospholipid was quenched by about 80% when the gold-nanoparticles are formed (found by exposing the particles to a detergent that removes the lipid layer). This quenching arose mostly from a quenching by the gold, with some contribution from self-quenching in the lipid bilayer. Finally, they determined that while the lipids could exchange from one gold nanoparticle to another, they were not removed from the gold nanoparticles when taken up by cells. This was established by mixing two types of gold nanoparticles labeled with different fluorescent lipids and observing energy transfer in solution and colocalization in cells.

In this work, the size and monodispersity of PCAuNPs was measured in water by Malvern Nano-S Dynamic Light Scattering (DLS). The size of the PCAuNPs was determined to be approximately 20 nm, consistent with previously published work (Figure 1) [51].

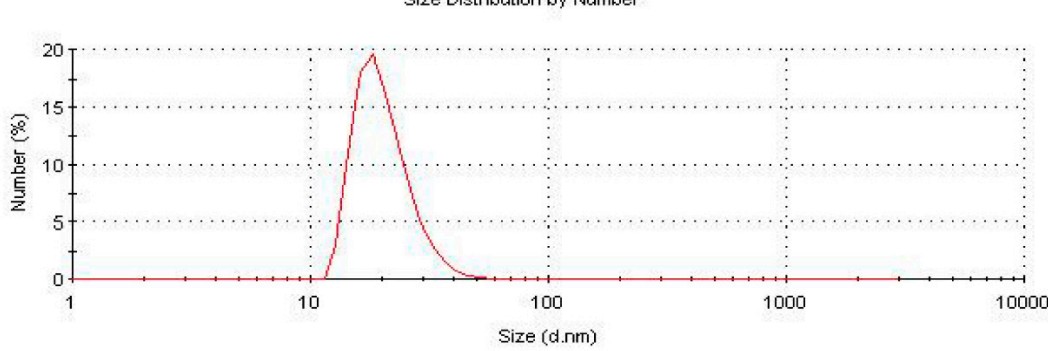

**Figure 1.** Size distribution of PCAUNPs in water from dynamic light scattering (DLS).

### 2.4. Variable Uptake of PCAuNPs in Cells

Four samples were required for each experiment. At 60% cell surface confluency, cell culture media was replaced with 1 mL of fresh media. Next, 100 µL of PCAuNPs was added to each sample. Samples were placed in a 37 °C, 5% $CO_2$ maintained incubator. Each sample was removed from the incubator after two hours had elapsed. Media was removed from each sample, and subsequently washed with 1X PBS. Then, 1 mL of fresh media was added to samples prior to being placed back into the 37 °C, 5% $CO_2$ maintained incubator. Each of the samples was removed from the incubator at different time points; after one, two, four, or twenty-four hours elapsed, respectively. Samples were washed at room temperature with 1X PBS and then exposed to 1 mL of 4% PFA at room temperature for ten minutes for cell fixation. Samples were then washed with 1X PBS for a final time and used for imaging.

### 2.5. Variable Exposure of PCAuNPs in Cells

Four samples were required for each experiment. At 60% cell surface confluency, cell culture media was replaced with 1 mL of fresh media. Next, 100 µL of PCAuNPs was added to each sample. Samples were placed in a 37 °C, 5% $CO_2$ maintained incubator. Each sample was removed from the incubator at different time points; one, two, four, or twenty-four hours after addition. Samples were washed at room temperature with 1X PBS, and then exposed to 1 mL of 4% paraformaldehyde (PFA) at

room temperature for ten minutes to fix the cells. Samples were washed with 1X PBS for a final time and used for imaging.

### 2.6. Immunofluorescent Labeling of Endocytic Compartments

Labeling of the various endocytic compartments was accomplished with antibodies specific for Rab5, Rab7, LAMP-1, and Rab11 protein endocytic markers. The primary antibodies used were monoclonal and originated from three different species (mouse, rabbit, and rat) to avoid potential crosslinking in experiments with multiple labels. Polyclonal secondary antibodies labeled with different fluorescent probes were added to bind with the primary antibodies. The fluorescent probes of the polyclonal antibodies were labelled and identified by the manufacturer as Cy3, Alexa Fluor 488 nm, Alexa Fluor 555 nm, and Alexa Fluor 647 nm.

### 2.7. PCAuNPs Co-Localization with One and Two Endocytic Compartments

PCAuNP co-localization with early, late, recycling, and lysosomal endosomes was studied. The combinations of probes used for immunofluorescent labelling of PCAuNPs with endocytic marked compartments were the following; 488 nm and Alexa Fluor 647 nm, 488 nm and Cy3, 488 nm and Alexa Fluor 555 nm. Samples were maintained in a 37 °C, 5% $CO_2$ incubator with 1 mL of media and 100 μL of a nanomaterial system. After two hours, samples were washed with 1X PBS and fixed with 1 mL of 4% PFA. Samples were then labelled with primary and secondary antibodies specific for an endocytic compartment. For three color experiments, samples were washed and labelled with primary and secondary antibodies specific for the second endocytic marker of interest.

### 2.8. Image Acquisition

Images of the cells in a sample were obtained with a Carl Zeiss 710 Laser Scanning Confocal Microscope with a 63 × 1.4 Numerical Aperture Oil DIC Plan-Apochromatic lens. ZEN 2011 was the software used to adjust and select parameters for image acquisition. Lasers used for image acquisition were Argon Ion for Fluorescein Isothiocyanite (FITC) configuration (488 nm excitation of the Alexa Fluor 488 nm probe), Solid State for Cyanine-3 (Cy3) Configuration (561 nm excitation of the Cy3 and Alexa Fluor 555 nm probes), and HeNe for Cyanine-5 (Cy5) Configuration (633 nm excitation of the Alexa Fluor 647 nm probe). Complementary differential interference contrast (DIC) was used for focusing. The detectors for each configuration were adjusted in such a way that the emission from each channel was unique to a specific wavelength range. In addition, as a control measure, samples labelled with one fluorescent probe, such as Cy3, were illuminated with the FITC configuration, and then with the Cy5 configuration separately, to ensure that potential cross talk was minimized. In addition, controls showed that there was little non-specific binding of secondary antibodies. The pixel dwell time was set to 3.15 μs with an average of two line scans, and the pinhole was set to 1 airy-unit (AU). A Zoom factor of 14 was used to achieve a pixel size of approximately 20 nanometers. Each image comprised a square of 512 × 512 pixels, resulting in a 10 × 10 micron image for image correlation spectroscopy analysis. For each experiment, 25–40 images were obtained; each image was recorded from a different cell in the sample.

### 2.9. Principles of Calculations of Image Correlation Functions

The theoretical foundation for calculating correlation functions from images was established with the introduction of image correlation spectroscopy by Petersen and co-workers [57]. The auto-correlation function can be calculated directly as the sum of the products of all the pair-wise intensities $i(x, y)$ and $i(x + \xi, y + \eta)$ as indicated by the following equation:

$$g(\xi, \eta) = \frac{\langle (i(x, y) - \langle i(x, y) \rangle)(i(x + \xi, y + \eta) - \langle i(x, y) \rangle) \rangle}{\langle i(x, y) \rangle^2} \tag{1}$$

where the angular brackets indicate averaging over all spatial coordinates, x and y.

For images of any size, this becomes computationally prohibitive, so the alternative is to recognize that the auto-correlation function is the reverse Fourier transform (FFT$^{-1}$] of the power spectrum of the intensities in the image and that the power spectrum can be calculated as the Fourier transform [FFT] of the image multiplied by its complex conjugate [57]. Thus, the following applies:

$$\widetilde{i}(\xi', \eta') = \text{FFT}[i(x, y)] \tag{2}$$

and

$$g(\xi, \eta) = \frac{\text{FFT}^{-1}\left[\widetilde{i}(\xi', \eta')\widetilde{i}^{\,*}(\xi', \eta')\right]}{i\langle(x, y)\rangle^2} \tag{3}$$

where the asterisk denotes the complex conjugate of the Fourier transform.

Correspondingly, the cross-correlation function between two images, say red, r, and green, g, is calculated as the reverse Fourier transform of the joint power spectrum [57], that is:

$$\widetilde{i}_r(\xi', \eta') = \text{FFT}[i_r(x, y)] \tag{4}$$

and

$$g_{r,g}(\xi, \eta) = \frac{\text{FFT}^{-1}\left[\widetilde{i}_r(\xi', \eta')\widetilde{i}_g^{\,*}(\xi', \eta')\right]}{\langle i_r(x, y)\rangle\langle i_g(x, y)\rangle} \tag{5}$$

where the asterisk denotes the complex conjugate of the Fourier transform. Importantly, it does not matter which image is used for the complex conjugate.

Once the auto- or cross-correlation functions have been calculated, they are fit to a two-dimensional Gaussian function to extract the amplitude of the correlation function, g(0,0). Thus,

$$g(\xi, \eta) = g(0, 0)\exp\left(\frac{-2(\xi^2 + \eta^2)}{w^2}\right) + g_0 \tag{6}$$

where $g_0$ is a factor that allows for incomplete decay of the correlation function and w is the width of the Gaussian function. Note that the use of a Gaussian function is determined by the use of a laser beam whose cross-sectional profile is Gaussian, thus w is the width of the laser beam for the auto-correlation function and the geometric mean of the two laser beams for the cross-correlation function.

In this study, the image correlation toolkit in ImageJ software was utilized and the following pieces of information from images were obtained: the normalized auto- and cross-correlation function amplitudes, the average image intensity, and the laser beam radius in microns. Regions of interest were selected to 256 × 256 of the 512 × 512 image to avoid dark regions of the image where there is no cell. Cross correlation of images from two channels from two *different* cells was used to determine the cross-correlation amplitude that can arise from uncorrelated fluctuations. This represents a lower limit for the significance of cross correlation amplitude arising from two channels from the *same* cell.

The first application of triple cross correlations to measure three-way interactions in solution was provided by Williamson and co-workers [58,59], who studied ribosomal assembly using components labeled with three distinct chromophores. The extension of this to study three-way interactions using three images has been developed theoretically by Anikovski and Petersen [unpublished]. As with the cross-correlation function of two images, it is possible to calculate a triple cross-correlation function of three images. In this case, the triple cross-correlation function is the reverse Fourier transform of a bispectrum, which in turn is the product of the Fourier transforms of two of the images multiplied with the complex conjugate of the Fourier transform of the third image [60]. Once again, it does not matter which image is used as the complex conjugate. Thus, we can calculate the bispectrum as

$$\widetilde{I}(\xi', \eta', \upsilon', \theta') = \widetilde{i}_r(\xi', \upsilon')\,\widetilde{i}_g(\eta', \theta')\,\widetilde{i}_b^{\,*}(-\xi', -\upsilon', -\eta', -\theta') \tag{7}$$

where the subscripts refer to the red, green, and blue image as an example.

The triple cross-correlation function in now calculated as

$$g(\xi,\ \eta,\ \upsilon,\ \theta) = \frac{\text{FFT}^{-1}\left[\widetilde{I}(\xi',\ \eta',\ \upsilon',\ \theta')\right]}{\langle i_r(x,y)\rangle\langle i_g(x,y)\rangle\langle i_b(x,y)\rangle} \tag{8}$$

Note that there are four lag parameters in this function because there are two between the first and the second image, and two between the second and the third image.

The amplitude of interest is found when all lag parameters approach zero, so the fitting function is now:

$$g(\xi,\ \eta,\ \upsilon,\ \theta) = g(0,\ 0,\ 0,\ 0)\exp\left(\frac{-(\xi^2 + \eta^2 + \upsilon^2 + \theta^2)}{w^2}\right) + g_0 \tag{9}$$

where w is given by the geometric mean of the width of the three laser beams used, i.e.,

$$w^2 = \sqrt[3]{w_r^2 w_g^2 w_b^2} \tag{10}$$

Using these equations, it is possible to develop the appropriate software to perform the calculations needed and extract the amplitudes used in the present work.

The detailed theory of the triple image cross-correlation function calculations (TRICCS) and their applications is planned to be published by Anikovski and Petersen in the near future.

For TRICCS measurements, images were analyzed using image correlation spectroscopy software written by Max Anikovsky. Parameters for the range of data points, triple cross correlation amplitude, laser beam radius, and the correlation value at infinite limits were estimated to obtain the auto-, cross-, and triple-cross correlation functions amplitudes, and intensities for single, double, and triple color images, in various combinations.

### 2.10. Data Analysis

The ICS analysis provides three key pieces of information: the average intensity, the amplitude of the correlation function, and the width of the correlation function, which is related to the laser beam width. In these experiments, these parameters were calculated from 25–40 images measured on different cells in the same sample. The experiments were repeated four times. The aggregate calculated parameters were averaged and compared by statistical analysis using ANOVA followed by Bonferonni post hoc tests for pairwise comparisons. The standard error of the means was calculated from the ratio of the standard deviation of the data divided by the square root of the number of degrees of freedom (here N−1).

## 3. Theory

### 3.1. Image Correlation Spectroscopy

Image correlation spectroscopy (ICS) is a quantitative method used to analyze the fluorescent intensity fluctuations in an image obtained from a laser scanning confocal microscope. This analysis allows one to retrieve: the average intensity of markers, the number of marked endosomes, the number of marked endosomes per square micron, the relative degree of aggregation of markers per endosome, and the extent of co-localization between two or three markers of interest on an endosome.

ICS is an intensity fluctuation analysis of images obtained using a laser scanning confocal microscope in which relative fluctuations in intensity, $\delta_n i(x,y)$, are used to calculate a normalized auto correlation function, $g(\alpha,\beta)$, where $\alpha$ and $\beta$ are lag distances in the images [57].

In the limit as α and β approach zero, the amplitude of the normalized auto correlation function, denoted by g(0,0), becomes the variance of the intensity fluctuations [57,61,62];

$$g(0,0) = \text{var}(\delta_n i(x,y)) \tag{11}$$

Whenever the intensity is proportional to the concentration of fluorescent probes (here PCAuNPs), the variance can be shown to be the inverse of the average number of PCAuNPs or clusters of PCAuNPs, defined as $\langle N \rangle$ in the observation region;

$$g(0,0) = \text{var}(\delta_n i(x,y)) = \frac{1}{\langle N \rangle} \tag{12}$$

The cross-sectional area of the observation volume is given by $\pi\omega^2$, where $\omega$ is the $e^{-2}$ beam radius. Thus, we define the average number of PCAuNP-containing compartments per square micron, the cluster density (CD), as:

$$CD = \frac{1}{g(0,0)\pi\omega^2} = \frac{\langle N_{PCAuNP} \rangle}{\pi\omega^2} \tag{13}$$

The average intensity is proportional to the number of PCAuNPs, $\langle i(x,y) \rangle = c\langle N_M \rangle$, where c is a proportionality constant reflecting spectroscopic and optical parameters such as emission collection efficiency, quantum yields, and molar adsorption coefficients [61]. We define the degree of aggregation, DA, as:

$$DA = \langle i(x,y) \rangle g(0,0) = c\frac{\langle N_{PCAuNP} \rangle}{\langle N \rangle} \tag{14}$$

### 3.2. Image Cross Correlation Spectroscopy

Image cross correlation spectroscopy (ICCS) is an extension of ICS that enables the analysis of images collected from a cell with PCAuNPs and one marker, each with a different color, say green (*g*) and red (*r*) [61].

Using ICCS we calculate the auto-correlation amplitudes, $g_g(0,0)$ and $g_r(0,0)$, and the cross-correlation amplitude $g_{gr}(0,0)$. The average number of clusters that contain both green and red species, $\langle N_{gr} \rangle$, can be estimated from the cross-correlation and individual auto-correlation amplitudes:

$$\langle N_{gr} \rangle = \frac{g_{gr}(0,0)}{g_g(0,0)g_r(0,0)} \tag{15}$$

The extent of co-localization can be represented as fractions; the fraction of one species that associates with the other

$$F(g|r) = \frac{\langle N_{gr} \rangle}{\langle N_g \rangle} = \frac{g_{gr}(0,0)}{g_r(0,0)} \tag{16}$$

$$F(r|g) = \frac{\langle N_{gr} \rangle}{\langle N_r \rangle} = \frac{g_{gr}(0,0)}{g_g(0,0)} \tag{17}$$

Equation (16) defines the fraction of green PCAuNP-containing compartments that also have red endocytic markers. Correspondingly, Equation (17) defines the fraction of red endocytic markers that also contain green PCAuNPs.

### 3.3. Triple Image Cross Correlation Spectroscopy (TRICCS)

Triple image cross correlation spectroscopy (TRICCS) is a further extension of ICCS that enables the analysis of images of three colored markers to estimate the extent of co-localization of all three species, green (*g*), red (*r*), and blue (*b*) (Supplemental Information). Here, we study the extent of co-localization of "green" PCAuNPs with "red" and "blue" endocytic markers.

The average number of compartments that contain green, red, and blue markers, $\langle N_{grb} \rangle$, can be estimated from the triple cross correlation amplitude and the individual auto correlation amplitudes [63]:

$$\langle N_{grb} \rangle = \frac{g_{grb}(0,0,0,0)}{g_g(0,0)g_r(0,0)g_b(0,0)} \tag{18}$$

The fraction of one distribution of species that contain two other species are defined by the following:

$$F(g|rb) = \frac{\langle N_{grb} \rangle}{\langle N_g \rangle} = \frac{g_{grb}(0,0,0,0)}{g_r(0,0)g_b(0,0)} \tag{19}$$

$$F(r|gb) = \frac{\langle N_{grb} \rangle}{\langle N_r \rangle} = \frac{g_{grb}(0,0,0,0)}{g_g(0,0)g_b(0,0)} \tag{20}$$

$$F(b|rg) = \frac{\langle N_{grb} \rangle}{\langle N_b \rangle} = \frac{g_{grb}(0,0,0,0)}{g_r(0,0)g_g(0,0)} \tag{21}$$

As an example, Equation (19) defines the fraction of PCAuNP-containing compartments that also contain red and blue markers and Equation (20) defines the fraction of compartments with one marker that also contain PCAuNPs and a second marker.

## 4. Results

### 4.1. Phospholipid-Coated Gold Nanoparticles Are Internalized

We have previously observed that PCAuNPs induce the formation of lamellar bodies in A549 cells [52]. Figure 2 compares the differential interference contrast (DIC) microscopy images and confocal microscopy images of A549 and C2C12 cells after exposure to PCAuNPs.

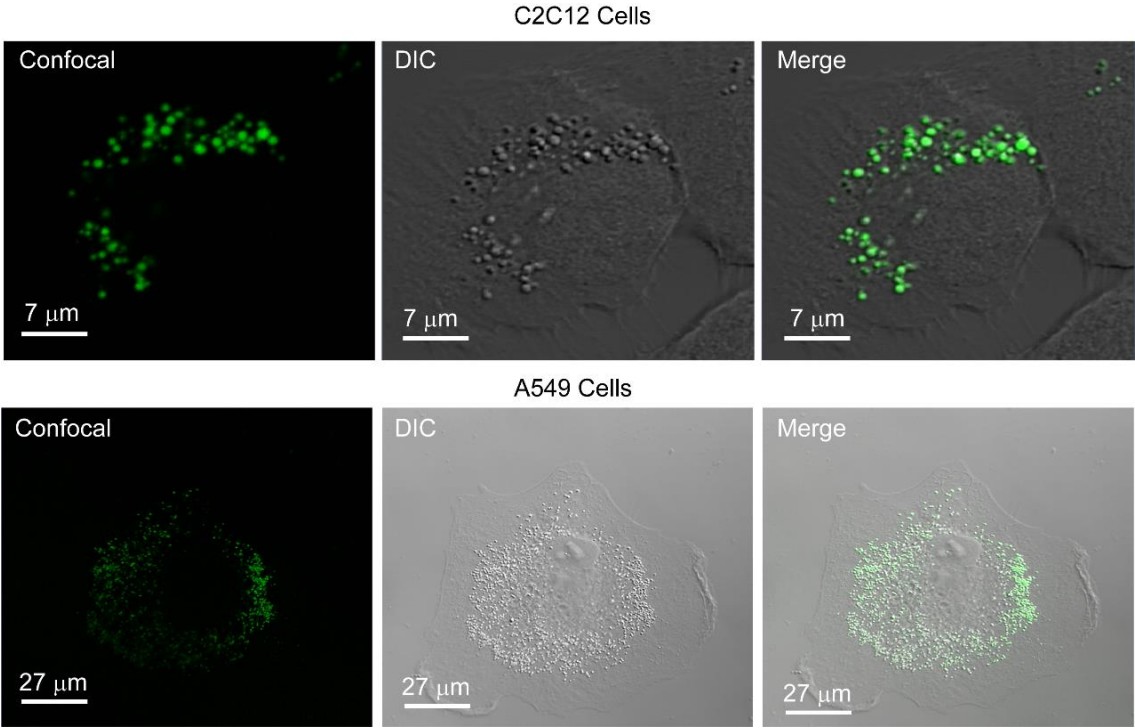

**Figure 2.** PCAuNPs internalize and form vesicular bodies in C2C12 and A549 cells.

In Figure 2, the images labelled "confocal" show the fluorescence clustering of phospholipid-coated gold nanoparticles. In the images labeled "DIC", the structural integrity of the cell is revealed and there are vesicular bodies present in the images. Lastly, the merge image reveals complete overlay of the fluorescent phospholipid-coated gold nanoparticles with vesicular bodies shown in the DIC image. The phospholipid-coated gold nanoparticles are internalized into the cells. This is consistent with previous work on A549 cells [51].

### 4.2. Endocytosis Occurs in C2C12 and A549 Cells and is Cell Type Specific

Based on previous studies in our lab, the PCAuNPs were found in acidic compartments [51]. In this work, we sought to clarify which acidic compartments are involved in the uptake of PCAuNPs by studying the extent of their co-localization with markers of compartments associated with the clathrin-mediated endocytic pathway.

We incubated cells with PCAuNPs (green) for two hours, followed by fixation and immunofluorescence labeling of either Rab5, Rab7, LAMP-1, and Rab11 compartments in two cell lines, C2C12 and A549 cells. Figure 3 shows contrast enhanced images of C2C12 cells with PCAuNPs (green) and markers for the early endosomes, late endosomes, lysosomes, and recycling endosomes using Rab5, Rab7, LAMP-1 and Rab11, respectively.

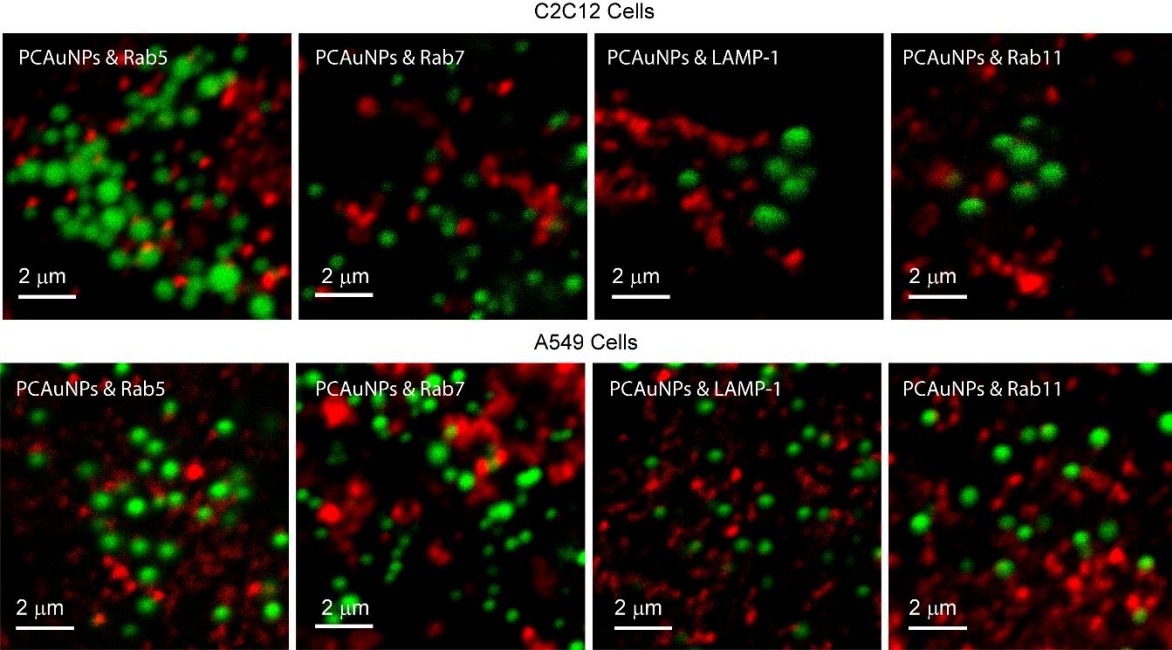

**Figure 3.** Endocytic marked compartments containing phospholipid-coated gold nanoparticles in C2C12 and A549 cells. Contrast enhanced fluorescence confocal microscopy images of C2C12 cells and A549 cells with PCAuNPs (green) and Rab5, Rab7, LAMP-1, and Rab11 markers (red) for early, late, lysosomes, and recycling endosomes, respectively.

The images in Figure 3 show that there appears to be little co-localization of PCAuNPs with clathrin-mediated endocytic markers in C2C12 and A549 cells due to the lack of "yellow" regions in the images. Quantitative determination of the extent of co-localization using image cross-correlation spectroscopy to obtain the fraction of PCAuNPs that associate with marked compartments is shown in Table 1. It is evident that there is some association of PCAuNPs with each of the four compartments.

**Table 1.** Co-localization of phospholipid-coated gold nanoparticles with endocytic marked compartments in cells.

|  | **C2C12** | **A549** |
|---|---|---|
| $\langle F(PCAuNPs|Rab5)\rangle$ | $0.21 \pm 0.03$ [⊣] | $0.64 \pm 0.05$ |
| $\langle F(PCAuNPs|Rab7)\rangle$ | $0.33 \pm 0.04$ | $0.38 \pm 0.08$ |
| $\langle F(PCAuNPs|LAMP-1)\rangle$ | $0.25 \pm 0.04$ | $0.57 \pm 0.08$ |
| $\langle F(PCAuNPs|Rab11)\rangle$ | $0.25 \pm 0.07$ | $0.49 \pm 0.04$ |

[⊣]. Standard error.

The fractions presented in Table 1 show that 21–33% of the PCAuNPs associate with each marked compartment in C2C12 cells while 38–64% of the PCAuNPs associate with each marked compartment in A549 cells. We can conclude that in both cell lines, these gold nanoparticles are found in the endocytic compartments. It is important to note that, because the markers for the various compartments are not unique [50], some of these compartments will have more than one marker associated with them. This point will be discussed further later.

Table 2 shows the fraction of individual marked compartments, or endosomes, that contain PCAuNPs.

**Table 2.** Co-localization of endocytic marked compartments with phospholipid-coated gold nanoparticles in cells

|  | **C2C12** | **A549** |
|---|---|---|
| $\langle F(Rab5|PCAuNPs)\rangle$ | $0.21 \pm 0.03$ [⊣] | $0.06 \pm 0.01$ |
| $\langle F(Rab7|PCAuNPs)\rangle$ | $0.17 \pm 0.02$ | $0.19 \pm 0.04$ |
| $\langle F(LAMP-1|PCAuNPs)\rangle$ | $0.22 \pm 0.04$ | $0.03 \pm 0.01$ |
| $\langle F(Rab11|PCAuNPs)\rangle$ | $0.18 \pm 0.04$ | $0.08 \pm 0.01$ |

[⊣]. Standard error.

The fractions presented in Table 2 show 17–20% of each endocytic marked compartment also contains PCAuNPs in C2C12 cells, while only 3–8% of the compartments contain PCAuNPs in A549 cells, with Rab7 compartments being an outlier with about 19% containing PCAuNPs.

In previous work, we observed that there was significant co-localization of Rab5 and Rab7 markers, of Rab5 and LAMP-1 markers, and of Rab7 and LAMP-1 markers, demonstrating that these markers are not uniquely defining their respective compartments [50]. We therefore sought to determine the fractions of PCAuNPs that associate with compartments with more than one marker. Accordingly, we incubated cells with phospholipid-coated gold nanoparticles followed by fixation and pairwise immunofluorescent labeling of Rab5 and Rab7; Rab5 and LAMP-1; Rab5 and Rab11; and Rab-7 and LAMP-1.

Figure 4 shows contrast enhanced images of C2C12 and A549 cells with PCAuNPs (green) and two pairs of endocytic markers (red and blue) as indicated in the figure.

The pink and purple spots in all four images reflect the co-localization of red with blue markers. Consistent with previous work, we observed that there are a few purple and pink spots in all images to indicate co-localization of Rab5 with Rab7, Rab11 and LAMP-1, and of Rab7 with LAMP-1 [50]. The association of PCAuNPs and Rab5 would be shown by a yellow color and the association of PCAuNPs with Rab7 and LAMP-1 would be shown by a cyan. Neither yellow nor cyan are present to a high extent. Co-localization of PCAuNPs with two markers would be observed as white regions.

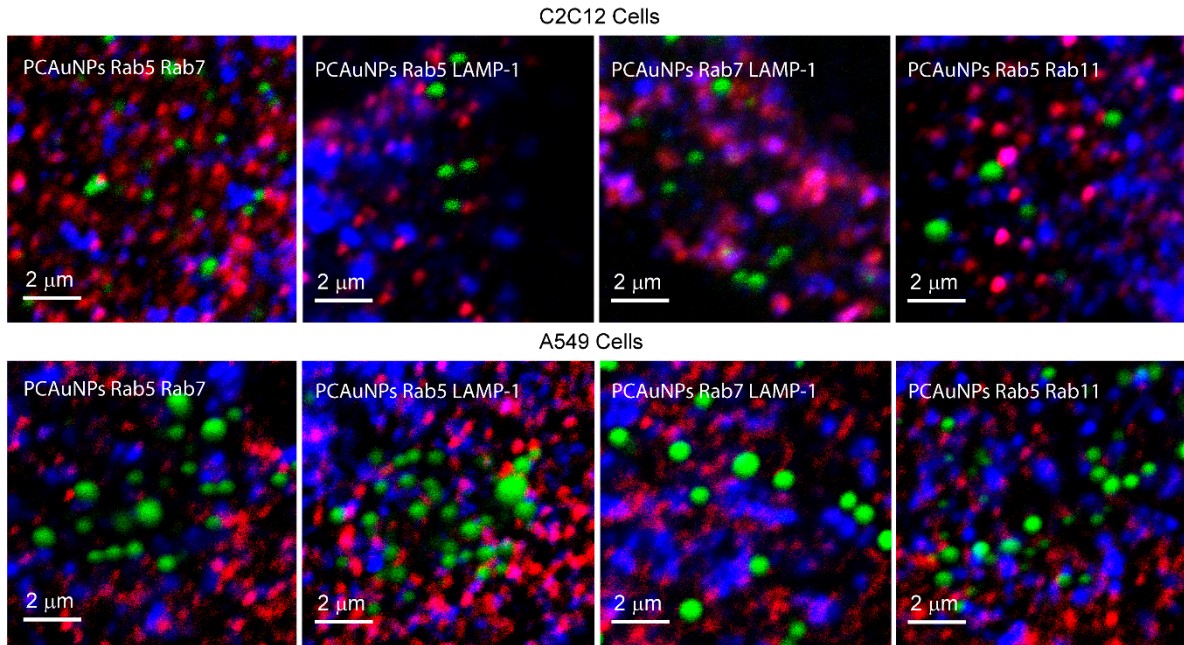

**Figure 4.** PCAuNPs with two endocytic marked compartments in C2C12 cells and A549 Cells.

To quantify the extent of co-localization from the images shown in Figure 4, triple image cross correlation spectroscopy was used to obtain the fractions of PCAuNPs that associate with two markers (Table 3).

**Table 3.** Fractions of PCAuNPs distribution that co-localize with compartments containing two markers.

|  | **C2C12** | **A549** |
| --- | --- | --- |
| $\langle F(\,PCAuNPs|Rab5\ Rab11\,)\rangle$ | 0.18 ± 0.05 [٦] | 0.33 ± 0.07 |
| $\langle F(\,PCAuNPs|Rab5\ Rab7\,)\rangle$ | 0.16 ± 0.05 | 0.25 ± 0.04 |
| $\langle F(\,PCAuNPs|Rab7\ LAMP-1\,)\rangle$ | 0.40 ± 0.08 | 0.24 ± 0.05 |
| $\langle F(\,PCAuNPs|Rab5\ LAMP-1\,)\rangle$ | 0.08 ± 0.02 | 0.39 ± 0.08 |

[٦]. Standard error.

It is clear that the PCAuNPs co-localize with compartments containing two markers in both cell types. Specifically, the fraction of PCAuNP-containing compartments that also contain two markers range from 8–40% in C2C12 cells and from 24−39% in A549 cells. These numbers are consistent with the fact that between 50 and 80% of these markers co-exist on the same endosome compartments [50].

Triple image cross correlation spectroscopy was also used to quantify the fraction of one marker that also co-localize with PCAuNPs and another marker (Table 4).

These fractions are consistent with the data in Table 2 and show that a significant, but small number of the various compartments contain PCAuNPs in C2C12 cells, while very few of these compartments contain PCAuNPs in A549 cells

**Table 4.** Fractions of marker distribution that co-localize with compartments containing PCAuNPs and one other marker.

|  | C2C12 | A549 |
|---|---|---|
| $\langle F(\,Rab11\,|PCAuNPs\,Rab5\,)\rangle$ | $0.23 \pm 0.07$ [†] | $0.04 \pm 0.01$ |
| $\langle F(\,Rab5\,|PCAuNPs\,Rab11\,)\rangle$ | $0.31 \pm 0.12$ | $0.03 \pm 0.01$ |
| $\langle F(\,Rab5|PCAuNPs\,Rab7\,)\rangle$ | $0.13 \pm 0.03$ | $0.02 \pm 0.00$ |
| $\langle F(\,Rab5|PCAuNPs\,LAMP-1\,)\rangle$ | $0.14 \pm 0.03$ | $0.05 \pm 0.01$ |
| $\langle F(\,Rab7|PCAuNPs\,Rab5\,)\rangle$ | $0.20 \pm 0.07$ | $0.05 \pm 0.01$ |
| $\langle F(\,Rab7|PCAuNPs\,LAMP-1\,)\rangle$ | $0.65 \pm 0.17$ | $0.01 \pm 0.00$ |
| $\langle F(\,LAMP-1\,|PCAuNPs\,Rab5\,)\rangle$ | $0.12 \pm 0.01$ | $0.02 \pm 0.00$ |
| $\langle F(\,LAMP-1\,|PCAuNPs\,Rab7\,)\rangle$ | $0.45 \pm 0.23$ | $0.06 \pm 0.01$ |

[†] Standard error.

*4.3. Uptake of PCAuNP Follows the Same Time Dependent Trend in Both Cell Lines—PCAuNP Aggregate to a Greater Extent in A549 Cells*

C2C12 and A549 cells were incubated with PCAuNPs for two hours, followed by rinsing, and re-incubation to monitor the processing of PCAuNPs in the cells. Figure 5 shows images of PCAuNPs after 1, 2, 4, and 24 h of re-incubation in A549 cells.

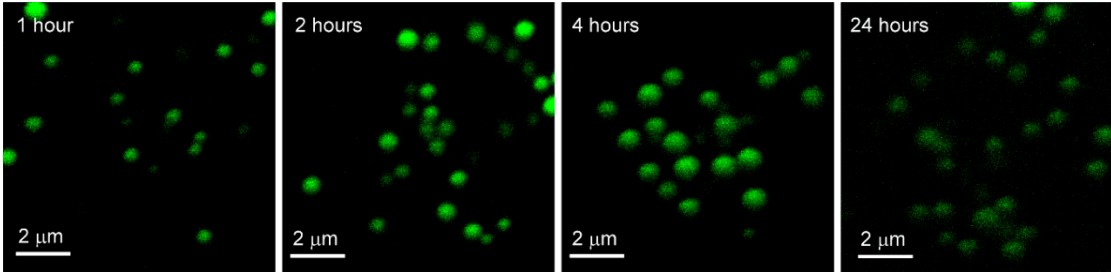

**Figure 5.** Images of phospholipid-coated gold nanoparticles after fixed exposure with variable uptake in A549 Cells. Confocal fluorescence microscopy images of A549 cells exposed to phospholipid-coated gold nanoparticles for 2 hours, washed, and re-incubated for 1, 2, 4 and 24 h respectively, then fixed.

In Figure 5, PCAuNPs are distributed in similar ways within the cytoplasm with the progression of time in A549 cells. However, the intensity of PCAuNPs appears to be less after 24 h has elapsed.

Image correlation spectroscopy (ICS) was used to quantify the intensity, cluster density, and degree of aggregation of PCAuNPs in C2C12 and A549 cells (Figure 6).

At short time frames, the intensity, or concentration, of gold nanoparticles in the cells are of the same order of magnitude, suggesting that the relative amount of gold nanoparticles internalized within two hours of initial exposure is comparable for the two cell types. Moreover, in the first four hours there is little change in the amount of PCAuNPs in the cells. However, after 24 h, the intensity decreased to less than half, consistent with the images shown in Figure 5 and suggesting that the PCAuNPs are being expelled or degraded (or at least the fluorescent lipids disappear). The cluster density (i.e., the number of compartments containing PCAuNPs per square micron) remains low for the first four hours, but increases significantly after 24 h, consistent with a dispersal of the PCAuNPs within the cells. Significantly, the cluster density is much lower in the A549 cells, suggesting a more aggregated state for the same number of PCAuNPs. Correspondingly, the degree of aggregation (i.e., the number of PCAuNPs per compartment) decreases rapidly, consistent again with a dispersal of the gold nanoparticles (or the fluorescent lipids) with time. Again, the degree of aggregation is much greater throughout in the A549 cells.

In summary, it appears that both cells types take up comparable numbers of gold nanoparticles, but that these are distributed in fewer and larger clusters (compartments) in the A549 cells. This is consistent with the observation from the cross-correlation experiments, that there are fewer compartments in the A549 cells that contain PCAuNPs.

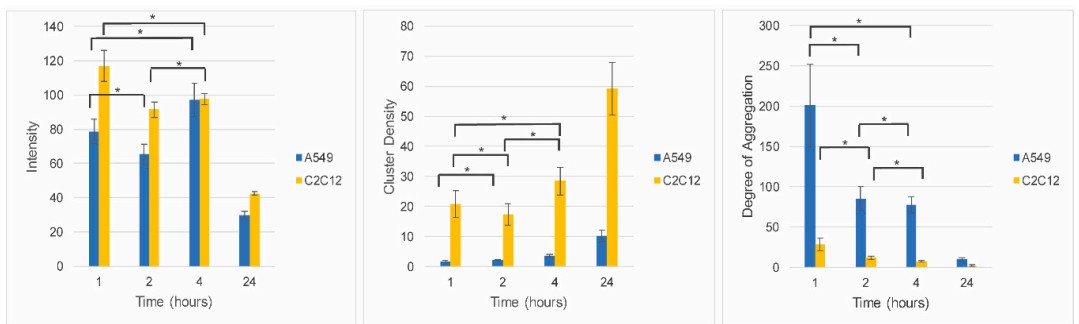

**Figure 6.** Fixed exposure, variable processing of phospholipid-coated gold nanoparticles in C2C12 cells and A549 cells. Image correlation spectroscopy data corresponding to A549 cells (**blue bars**) and C2C12 cells (**yellow bars**) exposed to PCAuNPs for 1, 2, 4, and 24 h. Error bars indicate standard error. ANOVA followed by Bonferroni post-hoc tests were performed for pairwise comparisons. There is no statistically significant change between the time points indicated by the * where $p > 0.0125$. Otherwise, all pairwise comparisons are statistically different and have $p \leq 0.0125$.

### 4.4. Phospholipid-Coated Gold Nanoparticles Aggregate Significantly with Continuous Exposure in A549 Cells

In the previous section, we observed that the gold nanoparticles taken up initially will eventually disperse into more, but smaller clusters. We now examine what happens with continuous exposure of the cells to the PCAuNPs over a similar time period.

The images in Figure 7 correspond to one confocal fluorescence microscopy image of one cell from each of the four samples prepared in the experiment, in which A549 cells were exposed to phospholipid-coated gold nanoparticles for 1, 2, 4 and 24 h, prior to fixation.

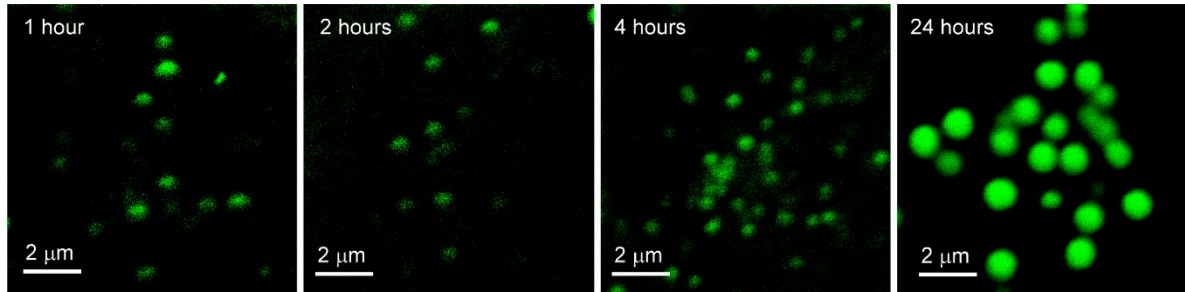

**Figure 7.** Images of phospholipid-coated gold nanoparticles after variable exposure with fixed uptake in A549 Cells. Confocal fluorescence microscopy images of A549 cells exposed to phospholipid-coated gold nanoparticles for 1, 2, 4 and 24 h, washed, then fixed.

The images show that there is a gradual increase in intensity and size of PCAuNP compartments over the 24 h.

Table 5 shows the averages of the auto correlation amplitude, average fitted laser beam width, and average intensity from a data set of 20–30 images of A549 cells exposed to phospholipid-coated gold nanoparticles for 1, 2, 4, and 24 h.

**Table 5.** Averages of ICS Parameters for PCAuNPs after variable exposure with fixed uptake in A549 cells.

| Time (Hours) | $\langle g(0,0) \rangle$ | $\langle \omega \ (\mu m) \rangle$ | $\langle I_{avg.} \rangle$ |
|:---:|:---:|:---:|:---:|
| 1 | 0.21 ± 0.04 [1] | 0.41 ± 0.01 | 82.8 ± 8.3 |
| 2 | 0.27 ± 0.07 | 0.39 ± 0.02 | 131 ± 15 |
| 4 | 0.65 ± 0.17 | 0.38 ± 0.01 | 138 ± 10 |
| 24 | 1.61 ± 0.18 | 0.48 ± 0.01 | 527 ± 43 |

[1] Standard error.

We observed an increasing trend of the auto correlation amplitude (g(0,0)) with time. The fit for the laser beam width ranges between 0.38–0.48 microns; this is consistent with the images shown in Figure 7 as larger size clusters were observed after 24 h. In previous work, we characterized the sizes of clathrin-mediated endocytic markers to be approximately 0.40 microns [50]. Thus, simply from this observation, one can assume phospholipid-coated gold nanoparticles may be contained in endocytic compartments. Additionally, the average intensity of phospholipid-coated gold nanoparticles increases with time of exposure in A549 cells, consistent with the images presented in Figure 7.

Table 6 shows the parameters derived from the data in Table 5 corresponding to the average number of clusters containing phospholipid-coated gold nanoparticles, the average number of clusters containing phospholipid-coated gold nanoparticles per square micron, and the average relative degree of aggregation for phospholipid-coated gold nanoparticles per cluster; calculated for a set of 20–30 images per sample.

**Table 6.** Averages of derived parameters for PCAuNPs undergoing variable exposure with fixed uptake in A549 cells.

| Time (Hours) | $\langle N_{AuPCNP} \rangle$ | $\langle CD \rangle$ | $\langle DA \rangle$ |
|:---:|:---:|:---:|:---:|
| 1 | 11.4 ± 2.1 [1] | 21.4 ± 3.6 | 16.4 ± 4.6 |
| 2 | 6.96 ± 1.33 | 14.8 ± 2.5 | 40.2 ± 15.8 |
| 4 | 3.73 ± 0.62 | 8.91 ± 1.70 | 71.2 ± 15.8 |
| 24 | 0.93 ± 0.16 | 1.49 ± 0.38 | 700 ± 49 |

[1] Standard error.

The data in Table 6 suggest that the number of clusters containing phospholipid-coated gold nanoparticles in the observation region of the beam as well as the cluster density decreases with time with a corresponding increase in the relative degree of aggregation. Thus, as more and more phospholipid-coated gold nanoparticles are taken up with ongoing exposure, they aggregate into fewer clusters that each contain many more gold nanoparticles. This is in stark contrast to the decomposition seen after a short exposure, suggesting that the uptake of new PCAuNPs is faster than their processing and decomposition.

## 5. Discussion

Our objective was to address three questions about the uptake of phospholipid-coated gold nanoparticles: (1) Are they internalized by clathrin-mediated endocytosis; (2) Does the process depend on cell type, and (3) What happens to them with time?

The images in Figure 2 show that the PCAuNPs are internal to the cells and that they co-localize with intracellular compartments. The cross-correlation data in Table 1 unambiguously show that these PCAuNPs are internalized in both cell lines into compartments that also contain markers that have been identified with early, late, and recycling endosomes as well as lysosomes. This is consistent with the previous observation that they are found in acidic compartments [51] in A549 cells. It is

also clear from the data in Tables 1 and 2 that the gold nanoparticles are associated with one of these four compartments, but that only a small fraction of the endosome compartments contain these gold nanoparticles. The triple cross correlation data in Tables 3 and 4 confirm the presence of the gold nanoparticles in compartments with multiple markers, which is consistent with recent observations that these markers are not unique to particular endosomes [50]. Thus, we conclude that these PCAuNPs internalize via the clathrin-mediated endocytosis pathway in both cell lines.

Both the auto- and cross-correlation data suggest that even though the extent of uptake of gold nanoparticles is similar in C2C12 and A549 cells (Figure 6), their distribution within the cells differ. In the A549 cells, the PCAuNPs appear in a smaller fraction of the endosome compartments than in C2C12 cells (Tables 2 and 4), which is consistent with the cluster density in A549 cells being lower than in the C2C12 cells (Figure 6). Thus, there are more PCAuNPs present in fewer compartments and they are therefore in a more aggregated state in A549 cells. It is not clear why this difference exists, but A549 cells are derived from type II alveolar cells that are designed to recycle liposomes of dimensions comparable to these gold nanoparticles [64]. As they are coated with phospholipids, the cell may be processing them as if they are liposomes [65], perhaps into lamellar bodies.

After a short exposure to the gold nanoparticles, the A549 cells appear to dispose of them over a 24-hour period (Figure 6 and Table 5). The number of clusters decrease as does the extent of aggregation within them. This is consistent with the observation from the cross-correlation experiments (Tables 1 and 2) that some of the PCAuNPs associate with Rab11, which is used as a marker for the recycling endosomes. They may offer a mechanism whereby the PCAuNPs are expelled from the cells.

When the A549 cells are exposed continually to the PCAuNPs, they appear to accumulate faster than they can be disposed of, and they are concentrating in fewer and larger endosomal compartments (Table 6). This is consistent with a rapid uptake mechanism and a slow disposal mechanism for these particles.

We conclude that these types of phospholipid-coated gold nanoparticles can be useful for studying the uptake and fate of gold nanoparticles in general and further studies may provide more details on the degradation or disposal mechanisms. Moreover, as they are coated with a lipid membrane, they also provide a vehicle for incorporation of membrane proteins or peptides or other lipids that may either target them to specific receptors or provide a delivery mechanism for drugs that are lipid soluble [66–68].

**Supplementary Materials:** The following are available online at http://www.mdpi.com/2073-4352/9/10/544/s1.

**Author Contributions:** L.J.S. and N.O.P. designed this work. L.J.S. performed the research and analyzed the data. L.J.S. and N.O.P. wrote the manuscript.

**Funding:** This work was funded by a Natural Sciences and Engineering Research Council (NSERC) Discovery Grant to NOP.

**Conflicts of Interest:** The authors declare no conflicts of interests.

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
