# Peer review of "Internalization of Phospholipid-Coated Gold Nanoparticles"

_crystals, doi:10.3390/cryst9100544_

Round 1

Reviewer 1 Report

Authors have addressed most of the concerns raised. Still, i) the quality of images need to be improved; ii) adequate clarification that the core advances in terms of analysis is published elsewhere must be provided; iii) the full characerization of the nanomaterials must be presented in the main text.

Reviewer 2 Report

The manuscript is overall interesting and could be considered for publication, with some explanations:

Despite the previous published manuscripts, authors could clearly explain why A549 and C2C12 cell lines. The use of primary cell cultures would have reinforced the results. A paragraph about statistical analysis is lacking, although in figure 6 some comparisons were done (ANOVA followed a post-hoc test?). Furthermore, I would compare the data of the two different cell lines in the tables 1-4. How was standard error calculated? Was it the variability of spectroscopic measures on the same sample or did the authors perform several replicates in some repeated experiments? I would move the theoretical part in the methods section in the supplementary material, and enrich the Discussion section, which is very basic. How can the difference between the cell lines influence the spectroscopic measurements in different tissues?

Minor revisions:

The aim of the study should be affirmations and not questions in the abstract section.

Round 2

Reviewer 2 Report

Only a point to be corrected: when pairwise comparisons are done, in the presence of more than two groups ANOVA followed a post-hoc test (Bonferroni, Tukey, etc) should be done instead of t-student.
